# Aberrant Methylation of Somatostatin Receptor 2 Gene Is Initiated in Aged Gastric Mucosa Infected with *Helicobacter pylori* and Consequential Gene Silencing Is Associated with Establishment of Inflammatory Microenvironment In Vitro Study

**DOI:** 10.3390/cancers14246183

**Published:** 2022-12-14

**Authors:** Hee-Jin Kim, Jong-Lyul Park, Byoung-Ha Yoon, Keeok Haam, Haejeong Heo, Jong-Hwan Kim, Seon-Young Kim, Mirang Kim, Woo-Ho Kim, Sang-Il Lee, Kyu-Sang Song, Kwang-Sung Ahn, Yong Sung Kim

**Affiliations:** 1Aging Research Center, Korea Research Institute of Bioscience and Biotechnology (KRIBB), Daejeon 34141, Republic of Korea; 2Korea Bioinformatics Center, KRIBB, Daejeon 34113, Republic of Korea; 3Department of Functional Genomics, KRIBB School of Bioscience, Korea University of Science and Technology (UST), Daejeon 34113, Republic of Korea; 4Department of Pathology, Faculty of Medicine, Seoul National University, Seoul 03080, Republic of Korea; 5Departments of Surgery, College of Medicine, Chungnam National University, Daejeon 35015, Republic of Korea; 6Departments of Pathology, College of Medicine, Chungnam National University, Daejeon 35015, Republic of Korea; 7Functional Genomics Institute, PDXen Biosystems Co., Daejeon 34129, Republic of Korea

**Keywords:** SSTR2, gastric cancer, WGBS, hypermethylation, inflammation

## Abstract

**Simple Summary:**

Somatostatin receptor 2 (SSTR2) is a key regulator of gastric acid secretion in the gastric epithelium. We revealed that *SSTR2* promoter is hypermethylated in intestinal metaplasia, dysplasia, and gastric tumors, associating with gene silencing. We suggest that *SSTR2* promoter methylation is initiated in aged gastric mucosae infected with *Helicobacter pylori* and consequential *SSTR2* silencing promotes the establishment of inflammatory microenvironment via the intrinsic pathway, providing new insights into gastric carcinogenesis.

**Abstract:**

The loss-of-function variants are thought to be associated with inflammation in the stomach. We here aimed to evaluate the extent and role of methylation at the *SSTR2* promoter in inflammation and gastric tumor formation. A whole-genome bisulfite sequencing analysis revealed that the *SSTR2* promoter was significantly hypermethylated in gastric tumors, dysplasia, and intestinal metaplasia compared to non-tumor tissues from patients with gastric cancer. Using public data, we confirmed *SSTR2* promoter methylation in primary gastric tumors and intestinal metaplasia, and even aged gastric mucosae infected with *Helicobacter pylori*, suggesting that aberrant methylation is initiated in normal gastric mucosa. The loss-of-function of *SSTR2* in SNU638 cell-induced cell proliferation in vitro, while stable transfection of *SSTR2* in AGS and MKN74 cells inhibited cell proliferation and tumorigenesis in vitro and in vivo. As revealed by a comparison of target genes differentially expressed in these cells with hallmark molecular signatures, inflammation-related pathways were distinctly induced in *SSTR2*-KO SNU638 cell. By contrast, inflammation-related pathways were inhibited in AGS and MKN74 cells ectopically expressing *SSTR2*. Collectively, we propose that *SSTR2* silencing upon promoter methylation is initiated in aged gastric mucosae infected with *H. pylori* and promotes the establishment of an inflammatory microenvironment via the intrinsic pathway. These findings provide novel insights into the initiation of gastric carcinogenesis.

## 1. Introduction

Gastric cancer (GC) is the sixth most common cancer worldwide and third highest cause of cancer-related deaths [1]. Thus, the useful biomarkers that define the malignancy potential of primary gastric tumors, inform prognosis, and could assist in establishing new therapeutic and preventive strategies were urgently needed for this disease. Approximately 95% of GC are adenocarcinomas, subdivided by the Lauren histopathology system into intestinal-type (IGC) and diffuse-type GC [2]. Gastric carcinogenesis starts as superficial gastritis and chronic atrophic gastritis (AG) and progress to intestinal metaplasia (IM), dysplasia (DP), and finally, carcinoma [3], indicating that a better understanding of chronic inflammation as a potent driver of GC development and progression is necessary to overcome this disease.

Acute *H. pylori* infection suppresses gastric acid secretion in parietal cells, resulting in a loss of feedback inhibition by acid and a compensatory increase in gastrin production in the antral G cells. Moreover, chronic *H. pylori* infection give rise to parietal cell loss and reduced acid production, which stimulates the G cells to overexpress gastrin [4]. Furthermore, inflammatory cytokines induced by *H. pylori* infection itself trigger the antral G cells to release gastrin [5]. This combination of achlorhydria and hypergastrinemia drives chronic inflammation, leading to oxyntic atrophy and the development of potentially preneoplastic spasmolytic polypeptide-expressing metaplasia, with an eventual progression to gastric carcinoma [6], indicating that *H. pylori* infection is one of the strong risk factors for gastric cancer.

Along with *H. pylori* infection, aging is also another risk factor that increases the chance of developing gastric cancer [7]. During aging, gastric stem cells replicate and epigenetic mosaicism was generated in the cells through stochastic errors in DNA methylation maintenance. Then chronic inflammation by *H. pylori* infection accelerates this process by promoting stem cell replication for tissue repair or by working directly on the epigenome. Continued epigenetic mosaicism results in restricted differentiation in some stem cells, leading to stem cell exhaustion and a selective growth advantage in other stem cells, which finally leads to clonal expansion and local hyperproliferation [7]. Thus, the combination of aging and chronic inflammation contributes to GC development. However, it is still elusive which target gene is epigenetically altered in normal gastric gland by aging and chronic inflammation and decisively contributes to GC development.

Gastrin release from the antral G cells is inhibited by somatostatin (SST), which is secreted by D cells in the pyloric antrum in response to luminal acid. In the fundus, the release of SST by D cells in response to neurohumoral agents mediates the direct inhibition of gastric acid secretion by the parietal cells and indirect inhibition by reducing histamine release by enterochromaffin-like (ECL) cells. These effects are mediated via somatostatin receptor subtype 2 (SSTR2), a G protein-coupled receptor (GPCR) [8]. It is plausible that SST modulates the gastrin–ECL cell–parietal cell axis via SSTR2 [9,10], suggesting that perturbation of regulators of this axis may trigger chronic inflammation in the gastric epithelium. Previous studies revealed that *SSTR2* mRNA expression was controlled by epigenetic modifications of its promoter in various human cell lines [11] and *SSTR2* promoter was remarkably hypermethylated in CRC and GC tissues when compared with adjacent normal tissues [12,13]. However, it is still not clear in detail how *SSTR2* contributes to gastric carcinogenesis. On the other hand, recent studies have suggested that SSTR2 may be a therapeutic target in nasopharyngeal carcinoma associated with infection by the Epstein–Barr virus [14] and that the increased *SSTR2* expression induced by the epidrugs, such as the DNA methyltransferase inhibitor and the histone deacetylase inhibitor, may improve treatment strategies for patients with neuroendocrine tumors [15].

In the current study, we aimed to determine whether the promoter methylation of *SSTR2* gene is initiated in normal gastric gland in association with aging and *H. pylori* infection and whether the promoter methylation contributes to gastric carcinogenesis. Further, we examined whether the *SSTR2* silencing promotes the establishment of inflammatory microenvironment at the cellular level and tumorigenesis in the xenograft model. The study provides novel insights into the role of *SSTR2* in the initiation of gastric carcinogenesis.

## 2. Materials and Methods

### 2.1. Human GC Cell Lines

Ten GC cell lines, i.e., SNU216, SNU601, SNU638, SNU668, SNU719, AGS, KATOIII, MKN1, MKN45, and MKN74, were obtained from the Korean Cell Line Bank (http://cellbank.snu.ac.kr/main/index.html) and cultured in RPMI 1640 medium supplemented with 10% fetal bovine serum and 1% antibiotic–antimycotic solution (Invitrogen, Carlsbad, CA). All experiments were performed using mycoplasma-free cells. Cell line identification was confirmed annually by STR profiling cell authentication provided by the UT Southwestern Genomic Sequencing Core and compared to the ATCC cell line profiles.

### 2.2. Human Materials

Fresh specimens by endoscopic submucosal dissection (ESD) from three individuals with early gastric cancer and 124 frozen tumors paired with adjacent non-tumor tissues were provided by the Biobank of Chungnam National University Hospital (CNUH, Daejeon, Republic of Korea), a member of the Korea Biobank Network.

### 2.3. Laser Capture Microdissection (LCM)

LCM was performed for fresh untreated ESD specimens as described previously [13]. Briefly, specimens were embedded in Tissue-Tek OCT medium (Sakura, Tokyo, Japan), ten serial sections per specimen were cut using a microtome (Leica Biosystems, Deer Park, IL, USA), and transferred to PALM membrane 1.0 PEN slides (Zeiss Microimaging, Munich, Germany). Slides were then stained with hematoxylin and eosin, and coated with Liquid Cover Glass N (Carl Zeiss, Germany). Gastric mucosa (GM), IM, DP, and gastric tumor (GT) cells were delineated using PALM Robosoftware (Zeiss Microimaging), cut out, and collected into 0.5-mL adhesive-cap tubes using a PALM LCM System (Zeiss Microimaging). Genomic DNA of the captured cells was isolated using QIAamp DNA Micro Kit (QIAGEN, Valencia, CA, USA) and its concentration was quantified using PicoGreen dsDNA Quantitation Kit (Molecular Probes, Eugene, OR, USA).

### 2.4. Whole-Genome Bisulfite Sequencing (WGBS) Analysis

DNA (50 to 100 ng) from the LCM samples was treated with bisulfite using the EZ DNA Methylation Gold Kit (Zymo Research, Orange, CA, USA). WGBS libraries were prepared using the TruSeq DNA Methylation Library Prep Kit (Illumina, San Diego, CA, USA), according to the manufacturer’s protocol. Library quality was evaluated using an Agilent 2100 Bioanalyzer and High-Sensitivity DNA Kit (Agilent, CA, USA). Paired-end sequencing was performed using HiSeq X Ten sequencing instrument (Illumina), yielding sequencing reads approximately 150 bp in size. For data analysis, raw fastq files were trimmed using Trim Galore (version 0.6.5) (https://www.bioinformatics.babraham.ac.uk/projects/trim_galore/, accessed on 26 August 2019) with a cutoff of q30. Thereafter, the trimmed whole-genome sequence fastq files were aligned with the human reference genome 19 (hg19), and duplicate reads were removed using Bismark (version 0.22.3) [16]. After data preprocessing and mapping, Qualimap (version 2.2.1) [17] was used to determine the mapping rate, duplication rate, mean coverage, and median insert size. Bismark was also used to determine CpG methylation. For identification of differentially methylated regions (DMRs), methylKit (version 1.20.0) [18] in R packages was used, with default parameter settings.

### 2.5. Bisulfite Sequencing

Genomic DNA from GC cell lines or tissues was treated with sodium bisulfite using EZ DNA Methylation Gold Kit (Zymo Research). Bisulfite-modified DNA (100 ng) was amplified using polymerase chain reaction (PCR) in a 20-μL reaction volume with primer sets specific to the bisulfite sequencing region (Appendix A) encompassing 33 CpGs at the promoter regions of *SSTR2* gene. PCR products were cloned using pGEM-T Easy vector (Promega, Madison, WI, USA), and 10 clones were randomly chosen and sequenced. The methylation percentage for each sample was estimated as the ratio of methylated CpG dinucleotides to the total number of CpGs.

### 2.6. Pyrosequencing

Three CpG sites at *SSTR2* promoter were selected for the quantification of the extent of methylation. Bisulfite-modified DNA (100 ng) was PCR-amplified in a 20-μL reaction volume using MG 2× Taq Master Mix with Dye (MGmed, Seoul, Republic of Korea) to yield 119-bp product, using specific primers (Appendix A). PCR was performed with an initial melting step of 95 °C for 10 min; followed by 40 cycles of 95 °C for 30 s, 58 °C for 30 s, and 72 °C for 30 s; with a final incubation at 72 °C for 7 min. Pyrosequencing was performed as described [19], using a sequencing primer (Appendix A) and PyroMark Q48 Advanced CpG Reagents (QIAGEN) with PyroMark Q48 Autoprep (QIAGEN).

### 2.7. Reverse-Transcription PCR (RT-PCR) and Real-Time Quantitative PCR (RT-qPCR)

Total RNA was isolated from GC cells or tissues using RNeasy Kit (QIAGEN), treated with DNase I (Promega), and reverse-transcribed using Superscript II reverse transcriptase (Invitrogen), as per the manufacturer’s instructions. Using primer set described in Appendix A, RT-PCR of *SSTR2* gene was performed, as follows: 94 °C for 5 min followed by 35 cycles of 94 °C for 30 s, 64 °C for 30 s, and 72 °C for 30 s; with a final cycle of 72 °C for 7 min. PCR products were analyzed on 1.5% agarose gel stained with ethidium bromide. RT-qPCR of *SSTR2* gene was performed using a C1000 Thermal Cycler (Bio-Rad, Hercules, CA, USA) and primer set indicated in Appendix A. For the reaction, cDNA (100 ng) was amplified under the same conditions as RT-PCR in the course of 45 amplification cycles with 2× SYBR Green Supermix (Bio-Rad). *β-actin* gene was amplified as a control. The relative amount of target mRNA was determined using comparative threshold cycle (Ct) method [20].

### 2.8. Cell Treatment with 5-aza-2-deoxycytidine (5-aza-dC) and Trichostatin A (TSA)

Cells were seeded at a density of 1 × 10^6^ per 100-mm dish, cultured for 24 or 72 h, and treated with 1 μM 5 of -aza-dC (Sigma-Aldrich, St. Louis, MO, USA) as a demethylating agent, and then with 0.5 μM of TSA (Sigma-Aldrich) as an inhibitor of histone deacetylase. After 2–5 d, cells were washed with PBS, and total RNA was isolated using RNeasy Kit (QIAGEN). The RT-qPCR analysis of *SSTR2* expression was performed as described in Section 2.7. Independent triplicate experiments were performed.

### 2.9. Chromatin Immunoprecipitation (ChIP)-PCR Assay

ChIP was performed with GC cells following a protocol from the Myers’ laboratory (http://hudsonalpha.org/myers-lab/protocols, accessed on 15 June 2021) with modifications. Briefly, the cells were fixed with 1% formaldehyde, lysed, and sonicated using a Covaris M220 (Covaris, Woburn, MA, USA). For ChIP-PCR analysis, the sonicated lysates were used by dividing the same amount into three tubes and a 10% input. Each of the samples were immunoprecipitated with normal Rabbit IgG (2 μg; Millipore; 12-370), anti-trimethyl-Histone H3 (Lys4) (2 μg; Milipore; 07-473) or anti-trimethyl-Histone H3 (Lys27) (5 μg, Millipore; 07-449) using 50 μL Dynabeads coupled with protein A and protein G (Invitrogen), respectively. Immunoprecipitated DNA was recovered using the QIAquick PCR Purification kit (Qiagen) and used to amplify three regions around TSS of *SSTR2*. The PCR products were analyzed on 1.5% agarose gels stained with GelRed (biotium). The primer sequences for RT-PCR are listed in Appendix A.

### 2.10. Immunohistochemical Analysis of Tissue Microarrays of Human GTs

To examine the SSTR2 protein levels, tissue microarray analysis of 450 resected GTs from Seoul National University Hospital (SNUH-20040GC; SuperBioChips) (http://www.tissue-array.co.kr/main.html, accessed on 30 March 2019) was performed. The tissue array blocks contained up to 60 cores in an 8 array, for a total of 450 cases for immunohistochemical staining. Each core included more than 10% tumors and internal controls, such as nonneoplastic GM or IM. Immunohistochemical analysis was performed using a Leica Bond-max automated immunostainer (Leica Micosystems, Newcastle, UK) with antibodies against SSTR2 (Abcam, Cambridge, UK) or chromogranin A (ImmunoStar, Hudson, WI), an endocrine cell marker, as recommended by the manufacturer. SSTR2 staining patterns were scored as 0 (negative), 1 (weakly positive), 2 (moderately positive), and 3 (strongly positive), which correspond to negative (0) and positive (1, 2, and 3) groups, accordingly.

### 2.11. Establishment of SSTR2-KO Cells using CRISPR/Cas9 System

To generate a single guide (sg) RNA for Cas9 targets for the deletion of the *SSTR2* coding sequence, three 20-nt sequences on target DNA preceding a 5′-NGG PAM sequence in exon 2 of the human genomic *SSTR2* locus were selected. Oligonucleotides were annealed to target sequences (5′-CACCG-3′, annealed upstream of 1–3 target sequences, and 5′-AAAC-3′, annealed upstream of the complementary 1–3 target sequences). Templates for sgRNA synthesis are listed in Appendix A. The sgRNAs were cloned into PX459 vector (addgene, Watertown, MA, USA) digested with *Bbs*I. The vector contains the human codon-optimized SpCas9 gene, a cloning site for insertion of sgRNAs, and a 2A-Puro resistance cassette for the selection of transfected cells. The clones were confirmed by DNA sequencing using a sequencing primer from the sequence of U6 promoter that drives the expression of sgRNA (Appendix A). SNU638 cells, where *SSTR2* is actively expressed, were seeded into 6-well plates at a density of 50 cells/well for 24 h before transfection. The cells were transfected using Lipofectamine 2000 (Thermo Fisher) and grown for 5 d in a medium containing 1 μg/mL puromycin (Sigma Aldrich). *SSTR2*-KO SNU638 cells were screened using T7E1 assay, and the targeted genomic site was sequenced using a sequencing primer (Appendix A). The loss-of-function of *SSTR2* in *SSTR2*-KO SNU638 cells was examined by using Western blot with rabbit monoclonal anti-SSTR2 antibodies (Abcam, ab134152) (Appendix A).

### 2.12. Establishment of SSTR2-OVER Cells using Lentivirus Infection

Full-length *SSTR2* cDNA was isolated from a human GC cell line (SNU638) using PCR with the primer set shown in Appendix A. The PCR product was digested with *Xba*I and *Not*I and cloned into pCDH-CMV-MCS-EF1-Puro vector (System Biosciences, Palo Alto, CA, USA). To confirm whether the vector contains unmutated wild-type gene, the construct was sequenced in both directions. For the lentivirus construction, HEK293T cells were seeded at a density of 3 × 10^5^ cells in 6-well plates and cultured for 24 h. The cells were then co-transfected with *SSTR2*-expression vector, that is the vector constructed based on pCDH-CMV-MCS-EF1-Puro, and MISSION Lentiviral Packaging Mix (Sigma Aldrich) using Lipofectamine 2000 (Thermo Fisher). Viral particles were collected and used to infect SNU719, AGS, MKN45, and MKN74 cells. Surviving *SSTR2*-expressing (*SSTR2*-OVER) cells were grown for 5 d with 1 μg/mL puromycin (Sigma Aldrich) and the *SSTR2* expression was examined using RT-PCR.

### 2.13. Cell Viability Assay

For the assay, *SSTR2*-KO and *SSTR2*-OVER cells were plated in a 96-well plate (SPL) at a density of 1 × 10^3^ cells/well. Cell numbers were monitored using EZ-Cytox Cell Viability Assay Kit (ITSBIO, Seoul, Korea) and a microplate reader (Molecular Devices, San Jose, CA, USA) at 450 nm. All cell viability assays were performed in triplicate.

### 2.14. Colony Forming Assay

For a colony formation assay in a monolayer culture, *SSTR2*-KO and *SSTR2*-OVER cells were plated at 1 × 10^3^ cells per well in 6-well plates. After 2 wks of incubation at 37 °C with G418, the cells were stained with crystal violet (0.5% crystal violet, 3.7% formaldehyde, 30% ethanol), photographed and counted.

### 2.15. Cell Migration Assay

Cell migration was examined in transwell chambers (6.5-mm diameter, 8-μm pore size; Corning, Corning, NY). *SSTR2*-KO and *SSTR2*-OVER cells (2 × 10^4^ per well) were added on the upper chamber, which contained 0.5 mL RPMI 1640 medium supplemented with 10% fetal bovine serum and 1% antibiotic–antimycotic solution (Invitrogen). Cells were incubated for 24 h at 37 °C and nonmigrating cells were removed with cotton swabs. Cells at the bottom of the membrane were stained with crystal violet and counted. Three microscopic fields for each well were chosen randomly and counted.

### 2.16. Wound-Healing Assay

*SSTR2*-KO and *SSTR2*-OVER cells were grown to confluency, and a wound was established using the ibidi Culture-Insert 2 well in u-Dish of 35mm. An amount of 70 μL of suspension cells were seeded into both sides of the ibidi Culture-Insert and incubated with complete medium at 37 °C and 5% CO_2_ overnight. We observed and photographed them at specific times (0 h, 6 h, 12 h, and 48 h) with an inverted microscope. The distances between the edges of the cells were measured using ImageJ software.

### 2.17. Tumorigenesis Assay in Nude Mice

We purchased 4-wk-old female BALB/c nude mice (Orient Bio, Gyeonggi-do, Republic of Korea) and maintained them in accordance with the guidelines and under approval of the Institutional Review Committee for the Animal Care and Use, KRIBB. *SSTR2*-OVER MKN74 and vector control-transfected cells were used in a tumorigenesis assay. For the tumorigenesis assay, cells (5 × 10^6^) in a Matrigel suspension (Corning, 354248) were subcutaneously injected into the flanks of 5-wk-old mice (5 mice per cell line). We measured tumor volume as described [21].

### 2.18. RNA Sequencing (RNA-seq)

RNA-seq was used to analyze the established *SSTR2*-KO and *SSTR2*-OVER cells. Total RNA from 1 × 10^7^ control cells, three *SSTR2*-OVER cells, and one *SSTR2*-KO cell were extracted using RNeasy Mini Kit (QIAGEN). RNA-seq libraries were prepared with three separate experiments per each cell type, using TruSeq RNA Sample Prep Kit (Illumina). Sequencing was performed using a Nextseq500 platform (Illumina) to generate 75-bp paired-end reads. Sequenced reads were mapped to a human reference genome (hg19) using the STAR alignment tool (version 2.5.1) [22], and gene expression was quantified using the count module in STAR. The edgeR package (version 3.12.1) was used to select differentially expressed genes (DEGs) based on RNA-seq count data. The value for each gene (in counts per million) was normalized by trimmed mean of M values, set to 1, and log2-transformed for further analysis. Pathway enrichment analysis of *SSTR2*-KO and *SSTR2*-OVER cells was performed using MSigDB_Hallmark_2020 [23].

### 2.19. Statistical Analysis

Human expression and methylation data are shown as the mean ± standard deviation (SD). For two-group comparisons, statistical analysis was performed using Student’s *t* test. A *p*-value below 0.05 was considered statistically significant. Statistical analysis of correlation between pairs of methylations, expressions, or methylations and expressions was performed using Pearson correlation coefficient. For in vitro assays, all data are representative of at least three separate experiments, and the results are expressed as the mean ± SD. For in xenograft assay, the results are expressed as the mean ± SE. All statistical analyses were performed using R statistical programming language (version 3.6.1).

## 3. Results

### 3.1. WGBS Analysis Show SSTR2 Promoter to Be Hypermethylated in Primary GT and Premalignant Tissues

WGBS libraries were prepared from bisulfite-treated LCM-DNA from three GM, three IM, two DP, and one GT samples from three patients (Figure 1A). The average read number per sample after Illumina HiSeq X Ten sequencing was 852,163,613, with the read length mapped per sample of 36,989,052,732 bp, indicating 80.1% mapping rate. The average duplicate rate was 34.21%. The sequencing depth of genome coverage was 11.79× on average (summarized in Appendix A). Because the sample number of DP and GT samples was small and these samples were at relatively late stages compared to IM in gastric carcinogenesis, we merged the data (i.e., for two DP samples from two patients and one GT sample from one patient) into a “GT” cell group for further analysis.

We first assessed methylation status at the promoters of *SSTR2* gene in LCM-DNA by visualizing the WGBS profiles using the UCSC Genome Browser (hg19). We confirmed that *SSTR2* promoters were hypermethylated in IM and GT cells compared to that of GM cells (Figure 1C), as shown in our previous study using MBD-seq and RRBS analysis [13]. To compare with the previous dataset from GM, IM, and GT tissues generated using Infinium HumanMethylation450 BeadChip (450K methylation, IL, USA) platform, we chose three CpG probes around the promoter region (Figure 1C). When WGBS data were visualized using the UCSC Genome Browser (hg19), the average methylation value of the three CpGs was significantly increased in IM (48.94) and GT (65.71) compared to that (6.56) of GM for *SSTR2* (Appendix A).

### 3.2. Abnormal Methylation of SSTR2 Promoter in Normal Mucosae Is Associated with H. pylori Infection and Aging

We next compared the methylation status of same CpGs in *SSTR2* promoter in 16 GM samples from GSE92863 [24] representing eight young and eight old volunteers with or without *H. pylori* infection. Methylation at three CpGs of *SSTR2* promoter was significantly increased only in *H. pylori*-positive old volunteers (Figure 1D, top). The methylation patterns in *H. pylori*-positive GM were also apparent in 108 IM tissues (Figure 1D, middle) from GSE103186 [25] and 372 GT tissues (Figure 1D, bottom) from The Cancer Genome Atlas (TCGA) portal. The data suggested that the abnormal hypermethylation of *SSTR2* promoter is initiated in GM from old patients infected with *H. pylori* and maintained in IM and GT tissues.

### 3.3. Abnormal Methylation of SSTR2 Promoter Is Associated with Gene Silencing in Clinical Tissues

We validated the promoter methylation and gene expression of *SSTR2* in clinical samples of the CNUH cohort. RT-PCR analysis revealed that *SSTR2* is expressed in non-tumors but is silenced or expressed at a low level in GT tissues tested (Figure 2A), with a similar expression pattern. Bisulfite sequencing of the same clinical samples revealed substantial differences in the average promoter methylation value between non-tumors (3.0%) and tumors (29.0%) (*p* = 0.0046) (Figure 2B), indicating a negative correlation between expression and promoter methylation in primary tumor tissues. When we used pyrosequencing and RT-qPCR to analyze 124 paired tissues from the CNUH cohort, the mean methylation value was significantly increased in tumors (19.05 ± 12.39%) compared with that (10.51 ± 12.15%) in non-tumors (*p* = 1.202 × 10^−6^) (Figure 2C, top, left) and the mean expression value was significantly decreased in tumors (11.55 ± 4.53) compared with that (14.31 ± 2.62) in non-tumors (*p* = 1.381 × 10^−7^) (Figure 2C, top, right), showing a negative correlation between methylation and expression (Figure 2C, bottom).

### 3.4. Immunohistochemical Analysis Reveals the Loss-of-Function of SSTR2 in Primary GT and Premalignant Tissues

In a healthy stomach, a large proportion of epithelial cells (neuroendocrine cells), including G, ECL, and parietal cells, expresses SSTR2 [26]. We reasoned that if *SSTR2* is silenced by epigenetic alteration in GC and premalignant lesions, we would observe its loss-of-function in the neuroendocrine cells. We examined SSTR2 levels in primary GT and premalignant tissues using tissue microarrays of normal GM and 432 GTs. Chromogranin A, a neuroendocrine cell marker, was detected in the antrum glands of G cells (N-PA) and the fundic glands of ECL and parietal cells (N-BD) of normal GM, as well as in IM, DP, and GT cells. On the other hand, immunohistochemical staining with anti-SSTR2 antibodies revealed that the protein is present in both the fundic and antrum glands of normal GM but not in IM, DP, and GT tissues (Figure 2D). We observed a loss of SSTR2 in 92.1% (398 of 432) of GTs tested (Figure 2E), indicating that SSTR2 expression was silenced in GTs and premalignant lesions. Clinicopathological analysis of clinical tissues revealed that the loss of SSTR2 was significantly higher in diffuse-type GC (95.5%, 168/176) than that in IGC (88.9%, 160/180) (*p* = 0.02) and that that in EGC (96.6%, 115/119) was significantly higher than that in AGC (90.4%, 283/313) (*p* = 0.03) (Appendix A).

### 3.5. Promoter Methylation of SSTR2 Is Associated with Gene Silencing in GC Cell Lines and Drug Treatment Restores SSTR2 Expression in GC Cell Lines

Next, we used RT-PCR and bisulfite sequencing to investigate the association of gene expression with promoter methylation of *SSTR2* in 10 GC cell lines. RT-PCR analysis revealed that *SSTR2* was strongly expressed in SNU668 cells, respectively, while it was weakly expressed in SNU638 or silenced in the remaining cell lines (Figure 3A). Bisulfite sequencing analysis revealed that the average methylation value at the *SSTR2* promoter was very low (6%) in SNU668 cells, and it was within the 31–93% range in the remaining cell lines (Figure 3B). This indicated that the *SSTR2* promoter methylation was associated with gene silencing in GC cell lines. To characterize histone marks associated with chromatin remodeling in *SSTR2*-silenced GC cells, we performed ChIP-PCR at the three regions around promoter region (Figure 3C). The MKN45 and AGS cells showed bivalent marks of H3K4me3 and H3K27me3 as indicative of an association with *SSTR2* silencing. On the contrary, SNU216 and SNU719 cells unusually showed only H3K4me3 signatures, which is an active histone mark. After the treatment with 5-aza-dC and/or TSA, *SSTR2* expression was restored in MKN45 cells by treatment with 5-aza-dC and 5-aza-dC/TSA and in AGS cells by treatment with 5-aza-dC, TSA, and 5-aza-dC/TSA, while it was restored in SNU216 and SNU719 cells only by treatment with 5-aza-dC/TSA (Figure 3D), suggesting that *SSTR2* expression in GC cell lines is regulated epigenetically by a different manner according to GC cell lines.

### 3.6. SSTR2 has Tumor Suppressor Activity In Vitro and In Vivo

To examine the possible activity of SSTR2, we established an *SSTR2*-KO SNU638 cell, in which the loss-of-function of *SSTR2* is induced by CRISPR/Cas9 system and *SSTR2*-OVER SNU719, AGS, MKN45, and MKN74 cells, in which *SSTR2* is ectopically expressed. Partial deletion of *SSTR2* coding sequence and the loss-of-function were confirmed in *SSTR2*-KO cells by PCR and Western blotting (Figure 4A, bottom). Ectopic *SSTR2* expression was confirmed in all *SSTR2*-OVER cell lines by RT-qPCR and Western blotting (Figure 4B). The loss-of-function of *SSTR2* significantly increased cell viability (Figure 4C, far left), colony formation (Figure 4D, left), migration (Figure 4E, left), and wound-healing activity (Figure 4F, top) than the control cells. On the contrary, ectopic *SSTR2* expression significantly decreased cell viability (Figure 4C, right four), colony formation (Figure 4D, middle and right), migration (Figure 4E, middle, and right), and wound-healing activity (Figure 4F, middle and bottom) than the control cells. These results suggest that *SSTR2* suppresses cell proliferation and migration signals in gastric cancer cells. We next examined the effect of ectopic *SSTR2* expression by a tumorigenesis assay. Nude mice were injected with *SSTR2*-OVER MKN74 cells or empty vector-transfected cells, sacrificed 34 d after injection, and their tumors were dissected and weighed (Figure 5). The control cells formed rapidly growing tumors, whereas *SSTR2*-OVER cells formed tumors that were much smaller (Figure 5B, C), suggesting that *SSTR2* has tumor suppressor activity and is a negative regulator of tumor growth both in vitro and in vivo.

### 3.7. Inflammatory Microenvironment Is Established by the Loss of SSTR2 In Vitro

To assess the molecular mechanism of how *SSTR2* contributes as tumor suppressor in vitro, we performed RNA-seq analysis for *SSTR2*-KO or *SSTR2*-OVER cells, followed by a pathway enrichment analysis based on the detected DEGs. From triplicate experiments, *SSTR2*-KO SNU638 cells expressed 16,149 transcripts on average (Appendix A), of which 2071 were selected as DEGs (log_2_|FC| > 0.5, *p* < 0.05) compared to control cells and divided into 976 downregulated and 1,095 upregulated genes. Next, we compared the DEGs with hallmark molecular signatures in the Molecular Signatures Database (MSigDB_Hallmark_2020) and revealed 18 pathways to be significantly induced and 14 pathways to be significantly inhibited (Appendix A). In particular, the induced pathways included hallmark molecular signatures, such as TNF-α signaling via NF-κB, interferon alpha response, interferon gamma response, TGF-β signaling, IL-2/STAT5 signaling, and the inflammatory response (Figure 6A). This result suggests that the loss-of-function of *SSTR2* may (induce) contribute to establish the inflammatory microenvironment for cancer-related inflammation. In other words, this data implies that the gain of *SSTR2* may inhibit the establishment of the inflammatory microenvironment. To compare the effect between the loss and the gain of function of *SSTR2,* we performed RNA-seq analysis for three *SSTR2*-OVER cells and the control cells, according to the same procedure for *SSTR2*-KO cells. *SSTR2*-OVER AGS cells expressed 15,682 transcripts on average, of which 241 DEGs (110 upregulated and 131 downregulated) were selected; for *SSTR2*-OVER MKN45 cells, 3017 DEGs (1,314 upregulated and 1,703 downregulated) from 16,157 transcripts; for *SSTR2*-OVER MKN74, 216 DEGs (160 upregulated and 56 downregulated) from 15,974 transcripts, respectively. When the downregulated DEGs from three *SSTR2*-OVER cells were pooled and compared with hallmark molecular signatures, the result revealed that 10 of 18 induced pathways in *SSTR2*-KO SNU638 cells were significantly inhibited in *SSTR2*-OVER cells (Figure 6A), especially in *SSTR2*-OVER AGS and MKN45 cells (Figure 6B), suggesting that *SSTR2* may inhibit cell signaling for cancer-related inflammation. In addition, some of the upregulated or downregulated DEGs in pathways in Figure 6B were validated *SSTR2*-KO or *SSTR2*-OVER cells using RT-qPCR analysis. The results showed that the expression levels of DEGs selected in pathways were well corresponded to those from RT-qPCR analysis (Figure 6C).

## 4. Discussion

In the current study, we aimed to determine the extent and role of *SSTR2* promoter methylation in inflammation and GT formation. We revealed that *SSTR2* promoter hypermethylation and a subsequent silencing of the gene may promote inflammation-related pathways in the tissue. These observations highlight the role of *SSTR2* in the initiation of gastric carcinogenesis.

In the current study, we focused on *SSTR2*, because it is a key regulator involved in the homeostasis of gastric acid secretion in the stomach [27]. Therein, the hormone SST functions as the main inhibitor of gastric exocrine (e.g., acid from parietal cell and pepsinogen from chief cell) and endocrine (e.g., gastrin from G cell, histamine in ECL cell, and ghrelin from Gr cell) secretion [28,29]. The biological activity of SST is mediated via five GPCRs, SSTR1–SSTR5 [30], of which the expression of *SSTR2* is the highest among the SSTR gene subtypes in the human stomach tissue. This indicates that SSTR2 is a key SST receptor and confirms the notion that it is the most ubiquitously expressed SSTR subtype in the gastrointestinal tract [31,32]. If *SSTR2* would be epigenetically silenced in GM, gastrin would be uncontrollably released from G cells in the pyloric mucosae, and gastric acid would be uncontrollably secreted by parietal cells in the oxyntic mucosae.

Although the question remains as to how DNA methylation is induced in GM, multiple factors, such as aging and chronic inflammation, have been implicated. Mechanistically, region-specific DNA methylation during aging is facilitated in a competitive manner by destabilization of the Polycomb repressive complex [33]. On the other hand, chronic inflammation triggered by *H. pylori* infection, but not *H. pylori* itself, plays a direct role in the induction of aberrant DNA methylation [34]. Of note, in one study, 61.8% of DNA hypermethylation detected in normal GM using 450K methylation array was caused by age-related methylation and 21.6% was caused by *H. pylori*-induced methylation [24]. Based on those 450K methylation data, we observed that methylation of *SSTR2* promoters in normal GM was associated with *H. pylori-* and age-related methylation. Notably, the methylation of *SSTR2* promoter was also detected in primary GTs of the TCGA and CNUH cohorts and premalignant IM tissues [23]. This indicates that the *SSTR2* methylation is initiated in normal GM by aging and chronic inflammation upon *H. pylori* infection.

It is well established that chronic inflammation is associated with increased risk of cancer [35]. Two molecular pathways linking inflammation and cancer are recognized: the extrinsic and intrinsic pathways [36]. In the former, inflammatory or infectious conditions augment the risk of developing cancer, as shown for *H. pylori* infection in the stomach [37]. Hence, the data presented herein suggest that aging and *H. pylori* infection may extrinsically contribute to the establishment of an inflammatory microenvironment in normal GM, resulting in aberrant methylation of *SSTR2* promoter. By contrast, the intrinsic pathway is activated by genetic events, such as activation of various types of oncogenes, chromosomal rearrangement or amplification, and epigenetic event, such as inactivation of tumor-suppressor genes through promote hypermethylation. The two pathways converge, resulting in the activation of transcription factors (e.g., NF-κB) that coordinate the production of inflammatory mediators and the activation of various leukocyte genes, creating a cancer-related inflammatory microenvironment [36]. However, no report on the role of SSTR2 in the intrinsic pathway has been published to date.

Here, we suggest for the first time that the intrinsic pathway is activated by epigenetic alteration of *SSTR2* in the neuroactive ligand–receptor interaction pathway. Once *SSTR2* is epigenetically altered in normal GM by aging and chronic inflammation upon *H. pylori* infection via the extrinsic pathway, the loss-of-function of SSTR2 could lead to the production of inflammatory mediators via the intrinsic pathway. We here demonstrated that the loss-of-function of SSTR2 induced inflammation-related and other pathways in vitro, ultimately promoting tumorigenesis. We also showed that ectopic *SSTR2* expression inhibited inflammation-related pathways and decreased cell survival and proliferation activity, suggesting that in vitro SSTR2 may intrinsically inhibit the production of inflammatory mediators and weaken inflammation. Hence, we suggest that the inflammation-related pathway is activated via the intrinsic pathway upon the loss-of-function of SSTR2, resulting in constitutive inflammatory conditions in GC cells, and increased cell survival and proliferation.

Taken together, we suggest that aging and *H. pylori* infection may induce aberrant methylation of *SSTR2* promoter in normal GM via the extrinsic pathway, while the loss-of-function of SSTR2 by aberrant promoter methylation may contribute to the establishment of the inflammatory microenvironment through the intrinsic pathway, promoting cancer-related inflammatory microenvironment. Further in vivo investigations are needed to elucidate whether the loss-of-function of SSTR2 recruits inflammatory cells via pathways that connect inflammation and cancer. Understanding the mechanisms that suppress inflammatory microenvironment, and thereby cancer formation, may be critical for the development of novel preventive and therapeutic strategies to treat GC.

## 5. Conclusions

In conclusion, we report that the promoter methylation of *SSTR2* gene is initiated in normal gastric gland in association with aging and *H. pylori* infection and that *SSTR2* silencing promotes the establishment of inflammatory microenvironment and tumor formation. This study provides novel insights into the role of *SSTR2* in the initiation of gastric carcinogenesis.

## Figures and Tables

**Figure 1 cancers-14-06183-f001:**
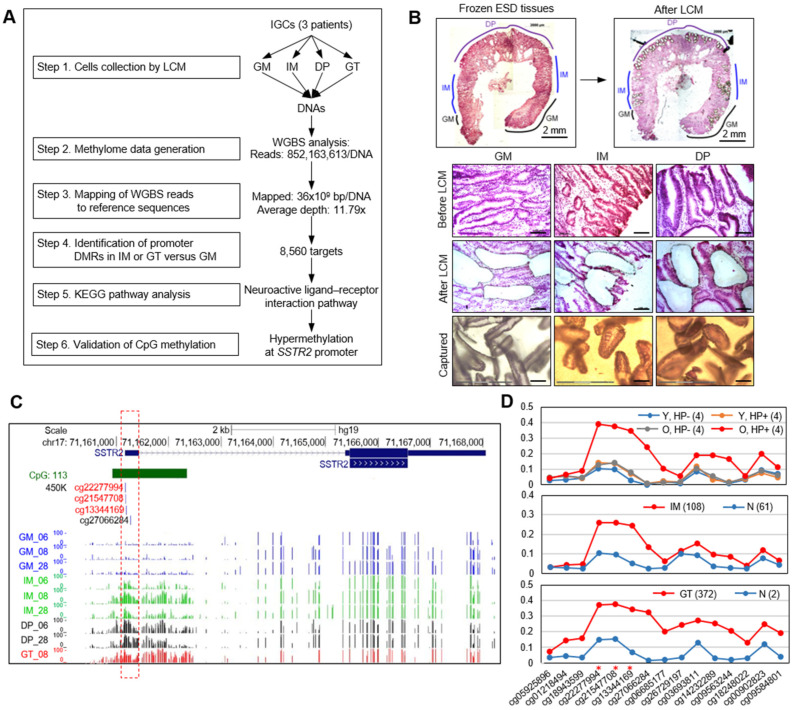
Methylome analysis for gastric tumors and the precancerous lesions. (**A**) Schematic diagram for WGBS analysis with clinical tissues. The six-step process was progressed for the initial methylome profiling and identification of promoter DMRs and a significant pathway associated with gastric carcinogenesis. (**B**) A representative example for isolation of gastric mucosa (GM), intestinal metaplasia (IM), and dysplasia (DP) cells from frozen ESD tissues of one patient (#06) with intestinal gastric cancer using LCM procedure. Upper panels show photos marked with gastric lesions before (left) and after (right) LCM. Lower panels also show photos enlarged before (top) and after (middle) LCM and captured tissues (bottom) for GM, IM, and DP cells, respectively. (**C**) Methylation profiles on *SSTR2* gene in GM, IM, DP, and GT cells. Top shows gene structure modified from UCSC Genome Browser (hg19): chr17: 71,160,089–71,168,292. Middle shows CpG probes that were selected for 450K methylation array. WGBS data of three patients (#06, #08, and #28) with EGC was visualized on UCSC genome browser as GM (blue colored), IM (green), DP (black), or GT (red) cells. The height of each colored vertical bar shows the extent of methylation within the range of 0 to 100 at single CpG site. Red dotted rectangle highlights hypermethylated promoter region, in which red-colored three CpG sites (cg22277994, cg21547708, and cg13344169) were chosen for purposes of comparison with the previous data for GM, IM, or GT tissues using 450K methylation array. (**D**) Detailed CpG methylation at promoter regions of *SSTR2* gene. The data was derived from 450K methylation data of GM (up), IM (middle), and GT (bottom). In case of GM, the 450K methylation data for four categories of each four samples (Yamashita et al., 2019) was used for this analysis: Y, HP—(blue line), young, and *H. pylori*-negative; Y, HP+ (yellow), young, and *H. pylori*-positive; O, HP—(gray), old, and *H. pylori*-negative; O, HP+ (red), old, and *H. pylori*-positive. In case of IM, the 450K methylation data for 108 IM and 61 normal tissues from GSE103186 were used, while for GTs, 372 GTs and two normal tissues from The Cancer Genome Atlas (TCGA) were used. At the top in each panel, the number of the tested sample was written in parenthesis. Bottom shows CpG sites as described in Figure 1C, and red asterisks indicate three CpG sites of interest as hypermethylated.

**Figure 2 cancers-14-06183-f002:**
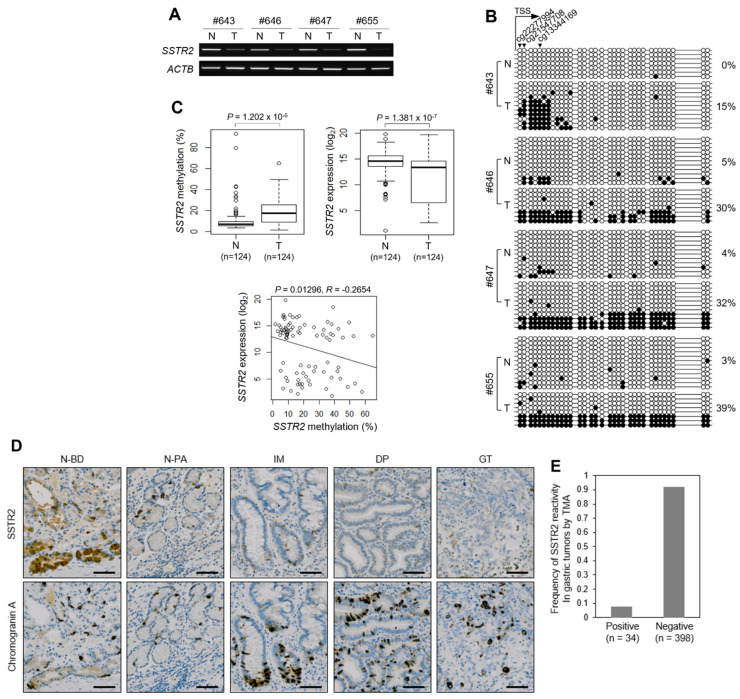
Promoter CpG methylation and expression for *SSTR2* in primary gastric tumors. (**A**) RT-PCR analysis in clinical tissues. *SSTR2* expression was examined in four paired gastric tumor tissues. *ACTB* was used as an internal control. (**B**) Bisulfite sequencing analysis at promoter regions. It was examined in the same clinical tissues used for RT-PCR analysis. Black and white circles indicate methylated and unmethylated CpG sites, respectively. Each row represents a single clone. Numbers on the right represent the mean percentage of CpG sites that were methylated in each sample tissue. Arrowheads in the top indicate three CpG sites, which are interested as the hypermethylated one and used for further pyrosequencing analysis. TSS, transcription start site. (**C**) Pyrosequencing (top, left), RT-qPCR (top, right) analysis, and correlation analysis between *SSTR2* methylation and expression (bottom). The analysis was performed for 124 paired non-tumor and gastric tumor tissues of CNUH cohort. Pyrosequencing was performed at three CpG sites shown in Figure 2B. Box plot analyses in panels show median, 25th, and 75th percentiles and outliers. (**D**) Tissue microarray analysis (TMA) with anti-SSTR2 in clinical tissues. TMA was performed with eight arrays containing a total of 432 gastric adenocarcinomas, including internal controls such as non-neoplastic GM or IM. Upper and lower panels show representative examples for immunostaining with anti-SSTR2 and anti-chromogranin A, respectively. Chromogranin A was used as a control marker for endocrine cell. N-BD, non-neoplastic gastric body; N-PA, non-neoplastic pyloric antrum. (**E**) Frequency of SSTR2 reactivity in human gastric adenocarcinoma by TMA.

**Figure 3 cancers-14-06183-f003:**
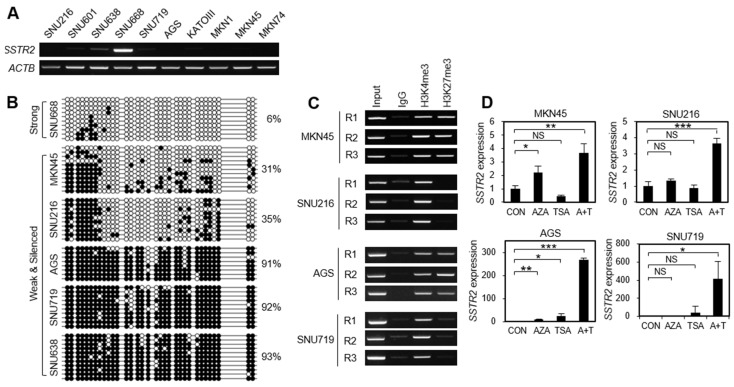
Promoter methylation and expression for *SSTR2* in GC cell lines. (**A**) RT-PCR analysis in 10 GC cell lines. *ACTB* was used as an internal control. (**B**) Bisulfite sequencing analysis for promoter regions of *SSTR2* gene. It was analyzed according to the same procedure of Figure 2B. Six GC cell lines were categorized into two groups (one Strong and five Weak and Silenced) based on level of *SSTR2* expression by RT-PCR analysis. (**C**) ChIP-PCR analysis at the promoter regions of *SSTR2*. Three regions were selected for this analysis: region 1 (−654 to −491 from TSS), region 2 (−69 to +124), and regions 3 (+244 to +421). H3K4me3 and H3K27me3 were used as active and repressed markers, respectively. IgG was used as a negative control. (**D**) Restoration of *SSTR2* mRNA expression after treatment with 5-aza-dC (AZA) and/or trichostatin A (TSA). The expression was examined by RT-qPCR analysis and normalized to *ACTB* expression in each sample. Each value is the mean ± SD of three independent experiments. * *p* < 0.05, ** *p* < 0.01, *** *p* < 0.001 versus untreated (CON) cells. NS, non-significant.

**Figure 4 cancers-14-06183-f004:**
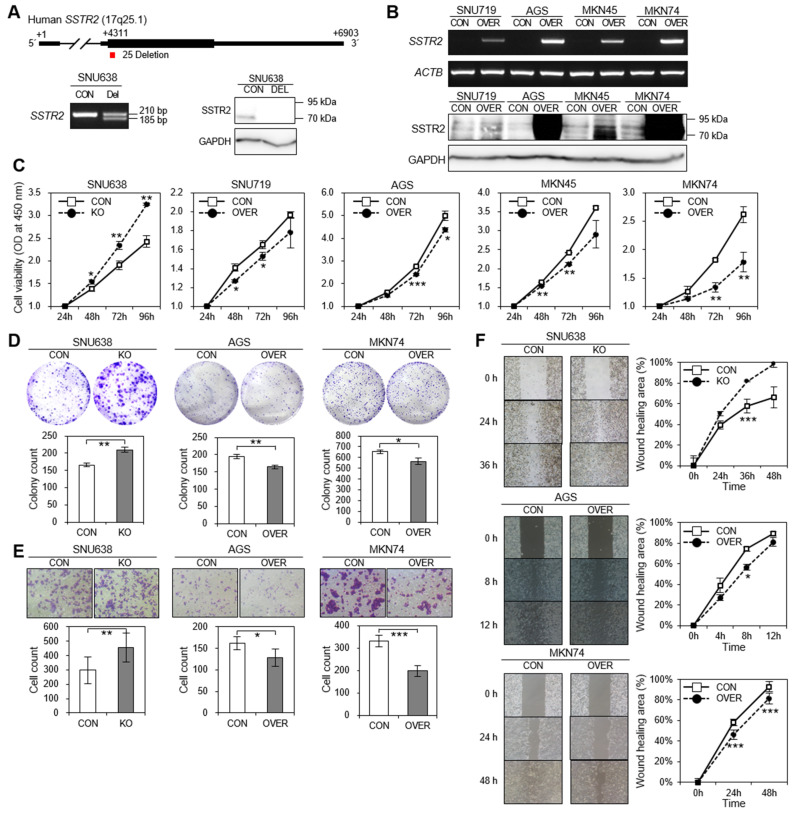
In vitro assays for *SSTR2*-KO and ectopic *SSTR2* expression. (**A**) Establishment of *SSTR2*-KO cells. Upper panel shows *SSTR2* gene structure and position of 25 bp deletion in coding sequence. PCR analysis shows 25 bp deletion (185 bp) in *SSTR2*-KO SNU638 cells by CRISPR/Cas9 system (left in bottom). SSTR2 expression was compared between control and *SSTR2*-KO cells by Western blot analysis (right in bottom). GAPDH was used as an internal control. (**B**) Establishment of ectopic *SSTR2*-expressing (*SSTR2*-OVER) cells. *SSTR2-*expression vector with a lentiviral was transfected into GC cell lines, such as SNU719, AGS, MKN45, and MKN74. *SSTR2* expression was examined in control and ectopic *SSTR2*-OVER cells by RT-PCR (upper) and by Western blot analysis (lower), respectively. *ACTB* or GAPDH was used as an internal control, respectively. (**C**) Cell viability assay. Relative viabilities of *SSTR2*-KO SNU638 or *SSTR2*-OVER SNU719, AGS, MKN45, and MKN74 cells over 4 days were measured using a CCK-8 kit and compared to the empty vector control (CON). (**D**) Colony formation assay. *SSTR2*-KO SNU638 or *SSTR2*-OVER AGS and MKN74 cells were plated on six-well plates at 1 × 10^3^ cells per well. After 2 weeks, colonies were stained with crystal violet and counted. (**E**) Migration assay. *SSTR2*-KO SNU638 or *SSTR2*-OVER AGS and MKN74 cells were plated on transwell chambers at 2 × 10^4^ cells per well. After 18–22 h, transwell membranes were stained with crystal violet and cells were counted. (**F**) Wound-healing assay. Gap closure assay with *SSTR2*-KO SNU638 or *SSTR2*-OVER AGS and MKN74 cells were performed at a density of 1 × 10^5^ cells. Scale bar, 100 µm. * *p* < 0.05, ** *p* < 0.01, *** *p* < 0.001 versus control (CON) cells.

**Figure 5 cancers-14-06183-f005:**
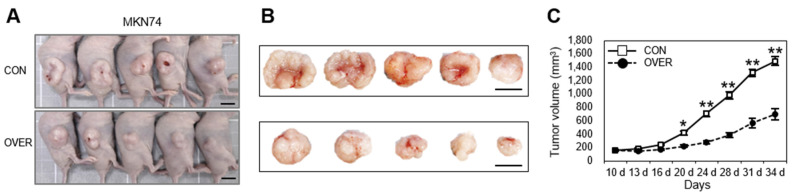
Xenograft assay with *SSTR2*-OVER MKN74 cells. (**A**) Mice were sacrificed at 34 d after injection of *SSTR2*-OVER MKN74 cells and control vector-transfected MKN74 cells into nude mice and tumor volumes were measured. Scale bar, 1 cm. (**B**) Photographs of tumors excised at 34 d after injection. (**C**) Tumor growth curves for *SSTR2*-OVER MKN74 cells and control MKN74 cells in nude mice. * *p* < 0.05, ** *p* < 0.01 versus control (CON) cells.

**Figure 6 cancers-14-06183-f006:**
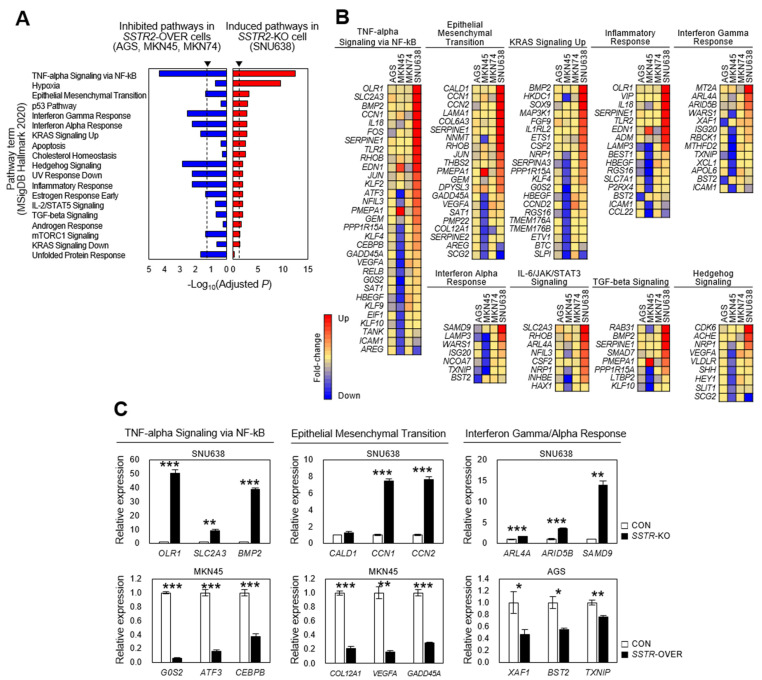
Pathway enrichment analysis in *SSTR2*-KO SNU638 and *SSTR2*-OVER SNU719, AGS, MKN45, and MKN74 cells. (**A**) DEGs in *SSTR2*-KO and *SSTR2*-OVER cells were identified compared to those of control cells through RNA-seq analysis, and pathway enrichment analysis was performed by comparison with gene sets of hallmark molecular signatures derived from the Molecular Signatures Database (MSigDB_Hallmark_2020). Induced pathways in *SSTR2*-KO cells were identified and compared to inhibited pathways, which were identified from pooled downregulated DEGs of three *SSTR2*-OVER cells. Red bars show induced pathways in *SSTR2*-KO SNU638 cells, while blue bars show commonly inhibited pathways in *SSTR2*-OVER AGS, MKN45, and MKN74 cells. The *x*-axis shows –Log_10_ (Adjusted *p* value) by Enrichment R analysis and the *y*-axis shows the pathway terms identified from MSigDB_Hallmark_2020, respectively. Arrowheads on top indicate a significant point, *p* = 0.05. (**B**) The expression intensity of genes in *SSTR2*-KO and *SSTR2*-OVER cells in inflammation and cancer-related pathways. For each pathway panel, top shows three *SSTR2*-OVER cells, such as AGS, MKN45, and MKN74, and one *SSTR2*-KO SNU638 cells and genes were listed in left. (**C**) Representative example of validation for top DEGs selected through the pathway enrichment analysis. This analysis was performed with *SSTR2*-KO or *SSTR2*-OVER cells and compared to control cells using RT-qPCR analysis. * *p* < 0.05, ** *p* < 0.01, *** *p* < 0.001 versus control (CON) cells.

## Data Availability

The raw data generated by WGBS and RNA-seq are available online at https://www.ncbi.nlm.nih.gov/geo/query/acc.cgi?acc=GSE166154 and GSE193993.

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
