# Peer review of "Aberrant Methylation of Somatostatin Receptor 2 Gene Is Initiated in Aged Gastric Mucosa Infected with Helicobacter pylori and Consequential Gene Silencing Is Associated with Establishment of Inflammatory Microenvironment In Vitro Study"

_cancers, 2022, doi:10.3390/cancers14246183_

Round 1
Reviewer 1 Report
This paper is well-written and interesting.
It should be written what WGBS stands for.
There are two related papers below, which should be included in the introduction or discussion.
1 “Epigenetic-Like Stimulation of Receptor Expression in SSTR2 Transfected HEK293 Cells as a New Therapeutic Strategy” Cancers 2022, 14(10), 2513
2 “SSTR2 in Nasopharyngeal Carcinoma: Relationship with Latent EBV Infection and Potential as a Therapeutic Target” Cancers 2021, 13(19), 4944
Reviewer 2 Report
The manuscript authored by Dr. Kim et al investigated the role of SSTR2 and its promoter hypermethylation. They found that SSTR2 hypermethylation was associated with H.pylori infection and plays a tumor suppressor role. Loss of SSTR2 promotes the establishment of an inflammatory microenvironment. Generally, the manuscript was well organized and well-written. The results are reliable and support the conclusion. The major concern is that DNA hypermethylation of SSTR2 and tumor suppressor role of SSTR2 have been reported in a variety of tumors, including gastric cancer. Other comments:
1. The interesting point of this manuscript is that the SSTR2 hypermethylation was associated with aged mucosa (not young) infected with H. Pylori, suggesting that H. Pylori may not be the major factor for DNA methylation, neither for age alone. It is suggested to carry out some experiments (in vitro) to do infection of gastric cells with H. pylori and check the DNA methylation change. Similarly, data is needed to check the DNA methylation level of SSTR2 in primary GC samples with vs without H.pylori infection and samples from older patients vs young patients (from your own samples , not the database to validate the results in Figure 1D.
2. Fig. 2, is there any correlation between SSTR2 methylation level and gene expression?
3. Fig.3 D, How did he DNA methylation level change after AZA/TSA/A+T treatments?
4. Fig.4 D, the data in AGS and MKN74 were not obvious. It needs to repeat the experiments using more cells and culture for longer time to let the colonies to grow.
5. It suggested to add IHC or western blot to show the levels of SSTR2 in tumors. If possible, IHC for validation of some findings in fig. 6 will improve the quality of the manuscript.
6. Fig.6. It needs to validate the top findings in this figure.
